# The Zoo of Isolated Neutron Stars

**Sergei B. Popov** 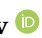

ICTP—Abdus Salam International Center for Theoretical Physics, Strada Costiera 11, I-34151 Trieste, Italy; sergepolar@gmail.com; Tel.: +39-040-2240-368

**Abstract:** In this brief review, I summarize our basic knowledge about different types of isolated neutron stars. I discuss radio pulsars, central compact objects in supernova remnants, magnetars, nearby cooling neutron stars (also known as the magnificent seven), and sources of fast radio bursts. Several scenarios of magneto-rotational evolution are presented. Recent observational data, such as the discovery of long-period radio pulsars, require the non-trivial evolution of magnetic fields, the spin periods of neutron stars, or both. In some detail, I discuss different models of magnetic field decay and interactions of young neutron stars with fallback matter.

**Keywords:** neutron stars; radio pulsars; magnetars; fast radio bursts

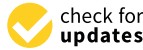



## 1. Introduction

Neutron stars (NSs) are very fascinating objects. The more we study them, the more interesting they seem to be.

There are numerous types of sources related to NSs. Many of them contain a NS as a member of a binary system. These are X-ray binaries with accretion on the compact object. The first discovery (but not identification) of such a source, Sco X-1, occurred in 1962 [1]. The first robust identification of a NS in an accreting binary, Cen X-3, occurred a few years later due to observations on-board the UHURU satellite [2].

Even as accreting objects, NSs in closed binary systems can appear as sources with very different properties. This is possible because binary evolution provides many possibilities to obtain systems with various parameters [3].

In low-mass X-ray binaries (LMXBs), mass transfer from a low-mass component spins up the NS. This can result in the formation of a millisecond radio pulsar (mPSR) [4]. The first mPSR was discovered in 1982 [5]. Spin up in a binary system was confirmed after identification of so-called transitional mPSRs [6]. Such NSs are characterized by very short spin periods ($\sim$1–10 ms) and low magnetic fields ($\sim 10^8$–$10^{10}$ G). The latter leads to large characteristic ages, $\tau_{\mathrm{ch}} = P/(2\dot{P}) \sim$ a few billion years, a and long life time.

Some PSRs (ordinary and mPSRs) are found in binary systems with other NSs [7]. There are several evolutionary channels which can result in the formation of such systems [8]. Some of these binary NSs are doomed to end their lives in a spectacular coalescence accompanied by a short gamma-ray burst (sGRB), a gravitational wave burst, and a kilonova [9].

NSs are usually found in binary systems due to their own activity (as in the case of PSRs), or due to their interaction with their companion (accretion or coalescence). However, recently, several candidates to inactive/non-interacting NSs in binaries have been reported on the basis of astrometric and spectroscopic data, see [10] and references therein for other proposed candidates.

Observations of binary systems provide a plethora of data on NSs. Still, in many cases, it is necessary to study isolated NSs, as in this case, properties of compact objects are not modified by the presence of a companion. In this review, I focus on different types of isolated (mostly young) NSs and their evolution.

## 2. Main Species in the Zoo

Observations in the whole range of electromagnetic waves, from radio to gamma, demonstrate very different phenomena related to isolated NSs. Various observational appearances and intrinsic properties (spin period, magnetic field, surface temperature, etc.) result in the classification of isolated NSs into several main types; see Figure 1. In this Section 1 briefly present descriptions of them.

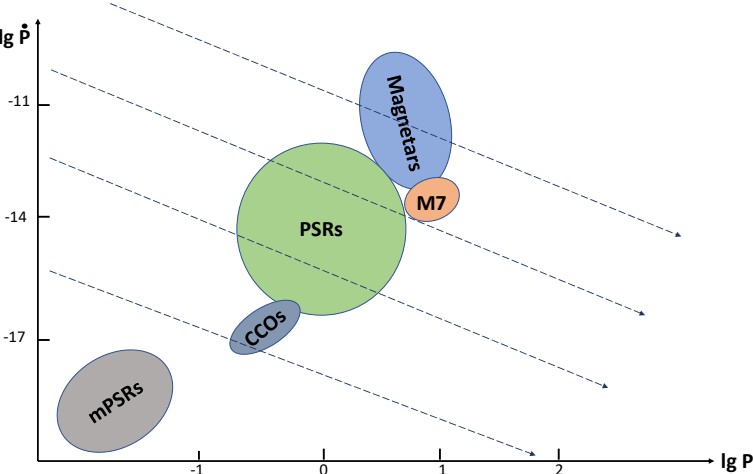

**Figure 1.** $P$–$\dot{P}$ diagram. The main types of young isolated NSs are shown together with recycled mPSRs (lower left part of the plot) and the simplest tracks corresponding to Equation (1) with a constant magnetic field. Scales on axes are approximate.

### 2.1. Radio Pulsars

Radio pulsars (PSRs) are the best-known (and the most numerous, referring to observed objects) type of isolated NS. They were discovered 55 years ago [11] through detection of their periodic radio pulses. The periodicity is due to the rotation of these compact objects. Spin periods cover the range of $P \sim 0.001$–100 s. Presently, the ATNF catalog[1] contains ~3000 of these sources [12]. Magnetic fields determined from the standard spin-down equation $B_{\mathrm{p}} = 3.2 \times 10^{19} \left( P\dot{P} \right)^{1/2}$ are in the range of ~$10^8$–$10^{14}$ G. These are dipolar fields on magnetic poles. On the equator, the field it twice as small: $B = B_{\mathrm{p}}/2$. Non-dipolar components might also exist, but their determination in the case of radio pulsars is not possible with any confidence, at least for now.

Low fields together with small spin periods correspond to old "recycled" mPSRs originating from LMXBs. In this section, we do not discuss them, focusing on young "normal" PSRs (in many respects, a group of so-called rotating radio transients (RRATs; see [13]) can be unified with PSRs, as they share main properties).

Radio emissions of PSRs are generated in magnetospheric processes which are not completely understood yet (see a review and references to early studies in [14]). Magnetospheric emissions might have a very wide spectrum. In some cases, e.g., the Crab pulsar, it is observed in the whole spectral range: from radio to gamma rays.

The energy reservoir is related to rotation of a PSR. This helps to relate key properties of a NS with observed parameters in a simplified magneto-dipole formula:

$$I\omega\dot{\omega} = \frac{2\mu^2\omega^4}{3c^3}. \tag{1}$$

Here, $I$ is the moment of inertia of a NS, $\omega = 2\pi/P$ is spin frequency, $\mu = BR_{\mathrm{NS}}^3$ is the magnetic moment, and $c$ is the speed of light. $R_{\mathrm{NS}}$ is the NS radius. Detailed calculations broadly confirm this equation [15]; however, alternative views also exist, e.g., [16].

Different studies indicate that the majority of young NSs pass through the stage of a PSR. Thus, the birth rate of PSRs is not much smaller than the birth rate of NSs in general.

The latter, in its turn, is not much smaller than the rate of core-collapse supernovae (CCSN), which is about $1/60 \, \text{yr}^{-1}$ [17]. All of these numbers are known with an uncertainty that is smaller than a factor of $\sim$2. Let us assume for a simple estimate that the birth rate of PSRs is $1/100 \, \text{yr}^{-1}$. Near the so-called "death line", where the number of observed pulsars in the $P$–$\dot{P}$ diagram drops (this might be related to a drop in efficiency of $ee^{-}$-pair production in the magnetosphere), a typical pulsar has $P \sim 2$ s and $\dot{P} \sim 3 \times 10^{-16}$ s/s. Then, its characteristic age is about $10^8$ yrs. Thus, the total number of PSRs in the galaxy is $\sim 10^6$. As radio emissions from PSRs are significantly beamed, just a small fraction (about 10%, [18]) of them can be observed even with infinite sensitivity. At the moment, the Five-hundred-meter Aperture Spherical radio Telescope (FAST) is the most sensitive instrument looking for new PSRs [19,20]. It is expected that SKA will allow us to detect most galactic pulsars that are potentially visible from Earth [21].

PSRs are characterized by large spatial velocities, $\sim$a few hundred km s$^{-1}$, which are about an order of magnitude larger than in the case of their progenitors [22]. Additional velocity, kick, is received by a NS during a SN explosion [23]. For young NSs, spatial velocity does not change significantly. Thus, there are hopes that the properties of velocity distributions of different subpopulations of isolated NSs can shed light on their origin and causes of diversity in their properties. However, it seems that velocities of different types of isolated NSs are similar to each other.

Young ages of normal PSRs are confirmed by their associations with supernova remnants (SNRs) [24] and spiral arms [25]. Some types of isolated NSs that will be discussed are even younger on average and are associated with SNRs by definition.

*2.2. Central Compact Objects*

Central compact objects in SNRs (CCOs for short) are young NSs observed due to their thermal emission inside supernova remnants. Known CCOs are not numerous, there about a dozen of them[2] [26,27]. However, their very small ages, about a few thousand years, point to a significant birth rate. Thus, this is a non-negligible subpopulation of isolated NSs.

CCOs can be an inhomogeneous group of sources united by their appearance inside SNRs and absence of any traces of radio pulsar activity. These objects are observed as soft X-ray sources with a thermal spectrum [28]. The origin of emissions is attributed to hot areas of a NS surface with a typical temperature of $\sim 10^6$ K. As the sources are young (ages are $\sim$few thousand years), the most obvious source of energy is residual heat. This assumption corresponds with the modeling of thermal evolution of NSs [29,30]. However, at least in a few cases, an additional source of energy is suspected.

Standard cooling is determined by neutrino emissions from the interior and photon emissions from the surface [31]. At the early stage of evolution, typically up to $10^4$–$10^5$ yrs, the neutrino emissions prevail. The neutrino emissivity strongly depends on the NS mass: low-mass objects can stay hot for a longer time. Still, residual heat can explain only temperatures of $\lesssim 3 \times 10^6$ K for ages $\lesssim 10^5$ yrs and $\lesssim 10^6$ K for ages $\lesssim 10^6$ yrs. Larger temperatures at young ages are typically explained by magnetic energy release, e.g., [32]. Measurable temperature in mature NSs (e.g., [33]) can be explained by chemical [34] or rotochemical heating [35].

In three CCOs, spin periods and their derivatives are measured due to X-ray flux variability. Periods $\sim$ 0.1–0.4 s and $\dot{P} \sim 10^{-17}$ s/s, according to the magneto-dipole formula, correspond to fields $\sim 10^{10}$–$10^{11}$ G. Because of their position in the $P$–$\dot{P}$ diagram, these CCOs are often called "antimagnetars". Still, some CCOs can be real magnetars.

Analysis of the light curve of PSR J1852+0040 in the SNR, Kes 79, allowed Shabaltas and Lai [36] to state that large pulse fractions observed in this source require a crustal field of the magnetar scale (see also [37]). Additional energy release due to field decay in the crust, or modification of the surface temperature distribution due to the influence of the magnetic field on heat transfer, might be responsible for the small emitting area in the PSR, J1852+0040, which may help to explain large pulsations of flux in the presence of the light bending effect in strong gravitational fields of compact objects. As no magnetospheric

activity was ever observed from this source, it was proposed in [36] that the NS is a "hidden" magnetar, i.e., the strong field is "screened" by the matter that falls back onto the NS soon after the SN explosion (see [38] and references therein to early papers regarding this scenario).

Another magnetar among CCOs is the famous NS inside the RCW 103 remnant. This source was discovered years ago at the Einstein observatory [39]. A prominent feature of the central source, which is its variability on a time scale of about a few years, was successfully described in the model of magnetic energy release in the crust [40], and it was suspected that the source can also belong to the class of "hidden" magnetars. However, it was soon demonstrated that it is not so hidden. Bright, short, high-energy bursts, quite similar to the bursts of soft gamma-ray repeaters, were detected from this source [41,42]. This clearly points toward the magnetar nature of the source. In addition, the 6.67 h spin period was measured for the NS in RCW103 [43]. Such long spin periods in young, presently non-accreting objects can be explained in a model where a strong magnetic field of the NS interacts with a fallback disc [44].

Despite that the majority of CCOs demonstrate only surface thermal emission, at least in a few cases can CCOs have an additional source of energy: its magnetic field, i.e., they can be related to magnetars.

*2.3. Magnetars*

A NS is called a magnetar if its observational appearance is mainly due to magnetic energy release. There are two main manifestations of these sources: high-energy bursts and surface thermal emissions. Often, magnetars are considered as the most extreme and interesting type of NSs. Their unusual properties manifest themselves in spectacular observational appearance, unfamiliar to other types of compact objects. During the hyperflare of SGR 1806-20 in 2004, the peak luminosity was above $10^{47}$ erg s$^{-1}$, and the total energy release in the event was $\sim 10^{46}$ erg [45]. This tremendous burst demonstrates that magnetic energy in NSs can reach very large values and can be rapidly released in a huge quantity.

It is not easy to say exactly when magnetars were discovered. The most reasonable approach may be to attribute it to the identification of the source, which was called, in 1979, "X-ray burster 0525.9-66.1" [46,47], and which is now mostly known as SGR 0525-66. Observations with gamma-ray detectors demonstrated the existence of subsequent powerful flares from the same source, including one giant flare with $L \gtrsim 10^{44}$ erg s$^{-1}$. A stable period of $\sim 8$ s was found, and the object was localized in a SNR in the Large Magellanic cloud (LMC). Now, the McGill catalog of magnetars[3] lists about 30 sources [48]. They belong to two main subclasses: anomalous X-ray pulsars (AXPs) and soft gamma-ray repeaters (SGRs). A recent review with a large bibliography dedicated specifically to these types of sources can be found in [49].

Initially, the division into AXPs and SGRs was very clear. Anomalous X-ray pulsars were characterized by relatively stable X-ray emissions with luminosity $\sim 10^{35}$ erg s$^{-1}$ (i.e., substantially smaller than in most accreting X-ray pulsars in binary systems); they had spin periods of about a few seconds, which were always increasing, and they did not have any optical or IR counterparts. Soft gamma-ray repeaters were characterized by intense bursts observed in the hard X-ray, in soft gamma-ray, or in both ranges, whereas AXPs did not show this type of activity. However, step by step, it became clear that AXPs and SGRs share similar properties. During active periods SGRs often resemble AXPs, and in 2002, Gavriil et al. demonstrated that well-known AXPs can have bursting activity identical to that of SGRs [50].

Typical spin periods of magnetars are about a few seconds. However, there are several important examples of outliers. A high-B young pulsar in the Crab-like plerion PSR J1846-0258, which started to demonstrate SGR-like flares, ref. [51] has a spin period of 0.327 s. Oppositely, the source 1E 161348-5055 in the SNR RCW 103 (which I described above) has a spin period of $\sim 6.67$ h.

The situation regarding magnetic fields is also not so univocal. "Classical" magnetars have fields of $\sim 10^{14}$–$10^{15}$ G. However, there are several so-called "low-field magnetars". Up to now, three objects have been reported (see a review in [52]). According to estimates, based on the usual magneto-dipole equation, these sources have dipole fields well below $10^{13}$ G, but phase-resolved spectroscopy has demonstrated the existence of proton cyclotron lines that indicate local surface fields of $\sim 10^{14}$–$10^{15}$ G [53,54]. These might be small-scale, non-dipolar components of the magnetic field.

The best definition of a magnetar involves magnetic field dissipation. That is, a magnetar is not just a NS with a large field, but a compact object with a magnetic energy release that dominates its luminosity for at least some period of time. The total energy budget can be roughly estimated as follows:

$$E_{\text{mag}} \sim \frac{4}{3}\pi R_{\text{NS}}^3 \left( \frac{B^2}{8\,\pi} \right) = 1.7 \times 10^{47}\, B_{15}^2 \text{ erg.} \tag{2}$$

Naive estimates of magnetar ages based on the characteristic age $\tau_{\text{ch}} \sim P/(2\dot{P})$ are not valid, as this simple equation is written for a constant field. However, the association of some magnetars with SNRs and their position in the galaxy [25] robustly confirm their young ages to be $\sim 10^3$–$10^5$ yrs. The young ages of magnetars indicate that an active period of magnetic energy dissipation does not last long.

Magnetars might constitute a significant fraction of young NSs. Many studies indicate their fraction to be $\lesssim 10\%$; see, e.g., [55] and references therein. However, at least one study suggests a much higher fraction of these objects: $\sim 40\%$ [56]. The question of the magnetar fraction is closely related to the problem of the formation of these NSs.

It is still unknown what defines if a newborn NSs is a magnetar. In one framework, it is necessary to have a strongly magnetized progenitor [57]. In another, the magnetic field is amplified by several orders of magnitude via a dynamo mechanism operating in a newborn NS [58,59].

In both scenarios, it is quite probable that evolution in binaries can play a role. For example, observations of the magnetic star, $\tau$ Sco, suggest that its magnetic field was substantially increased due to the coalescence of two main sequence stars in a binary, and with magnetic flux conservation, this star can become a magnetar in the future [60]. On the other hand, the evolution in binaries can result in significant spin-up of the stellar core, which might later make the dynamo mechanism efficient enough to produce a magnetar-scale magnetic field [61].

Rapid rotation, which is necessary for the production of large dipolar fields, with the dynamo mechanism [62] can be obtained in different ways. For example, a newborn NS can be spun-up due to fallback accretion [63].

Modeling of the magnetar evolution demonstrates [64,65] that after some time, for example, $\sim 10^4$–$10^5$ yrs, the rate of magnetic energy dissipation decreases, and all types of activity of a NS cease. Thus, magnetars become sources of a different type. One of their descendants are probably X-ray, dim, isolated NSs (XDINS), which are part of the magnificent seven (M7).

### 2.4. Magnificent Seven

The M7 (or XDINS) is a group of nearby ($\lesssim$ a few hundred pc; see [66]) young, isolated NSs observed due to their thermal surface emissions; see a review in [67,68]. The first member of this class of NSs was discovered in 1996 [69].

The first period in the history of studies of the M7 is dominated by results from the ROSAT satellite; see [70] for an early brief review and discussion (early ideas included, e.g., the possibility that these NSs can be accreting sources with decayed magnetic field [71]). Since then, many observations in different wavelengths have been obtained (in addition to X-rays, many sources are detected as dim optical sources with magnitudes of $\sim 26$–28 as well as in near-UV, near-IR, and in the deep upper limits of radio waves [72,73]).

Now, for all but one of these sources, spin periods and their derivatives are measured; see, e.g., Table 5 in [74]. The magneto-dipole formula provides an estimate of the magnetic field $\sim 10^{13}$–$10^{14}$ G. The observed thermal emissions can be either due to the residual heat, e.g., [75], or to some contribution from magnetic field decay [55].

Evolutionary, M7 might be the descendants of magnetars [55]. It was shown by population synthesis modeling that the population of the M7 originated mainly from the Gould Belt—the local ($\sim 500$ pc) star-forming structure [76]. As these NSs are relatively weak ($L \sim 10^{31}$–$10^{32}$ erg s$^{-1}$) and soft ($kT \lesssim 100$ eV), it is difficult to detect such sources at large distances, mostly due to interstellar absorption.

Despite intensive searches (e.g., [77] and references therein), very few NSs similar to the M7 have been found, and none of them ideally resemble the original seven sources. The first one was Calvera, a soft X-ray source high above the galactic plane [78]. However, later, it was shown that this NS has a short spin period (0.06 s), and in some other respects, it is different from the M7 sources [79]. The next one is 2XMM J104608.7-594306 [80]. Again, the spin period (18.6 ms) does not fit the M7 family [81]. Finally, the latest discovery is the source 4XMM J022141.5-735632 [82], which for this object, the spin period has not yet been reported.

There have been hopes that the eROSITA telescope can find more M7-like sources [83,84], but the survey program was terminated after only two years, and these hopes more or less disappeared.

## 3. Standard Evolution and Its Problems

It is convenient to discuss the evolution of young NSs in terms of $P$, $\dot{P}$, and $B$ and to illustrate it with the $P$–$\dot{P}$ diagram. In the simplest and the most standard way, the evolution is described by Equation (1) for $\mu =$ const. Then, NSs evolve in the $P$–$\dot{P}$ diagram along strait tracks, as shown in Figure 1.

The absence of significant field decay in normal PSRs is found in many papers, e.g., [85] (see, however, the next section). In addition, in PSRs, we do not see any evidence for the additional release of magnetic energy (the situation with magnetars is drastically different, of course).

Typically, in this standard approach, it is assumed that the initial spin periods are very short. Sometimes, authors can assume that the initial period is close to the limiting rotation (e.g., 1 ms). Sometimes, these initial spin periods are assumed to be close to the initial period of the Crab pulsar. Such assumptions were very popular, for example, in early models of binary population synthesis, e.g., [86]. Due to the gradual progress in our understanding of the initial parameters of NSs, such simplified approaches have been replaced by more advanced ones.

If $P_0 \ll P$ and the field is constant, then the real age of a pulsar is close to the characteristic age $\tau_{ch}$. Population synthesis studies and analysis of young NSs in SNRs with known ages indicate that typical initial periods of the majority of PSRs are of the order of 0.1 s [24,85,87]. Thus, for many standard PSRs with observed $P \sim 1$ s, the assumption of a small initial period can be acceptable. Still, in many cases, this does not work well and the results, e.g., when there is a significant discrepancy between the real age and $\tau_{ch}$.

The simplest model of magneto-rotational evolution nearly excludes links between different subpopulations: a CCO cannot become an M7-like object, and a magnetar cannot appear later in its life as a standard radio pulsar. This feature leads to an interesting controversy. The sum of birthrates of different subpopulations is larger than the rate of CCSN [88] (note, in that paper, the authors do not include CCOs in their calculations; with this subpopulation, the problem is even more severe). In addition, in the simplest model, it is difficult to explain the lack of magnetars with periods larger than $\sim 10$ s, e.g., [89]. This includes the absence of descendants of CCOs that might be visible at ages $\lesssim 10^6$ yrs when a SNR is already dissolved [90].

Thus, it is necessary to consider more complicated evolutionary paths.

## 4. Double Nature and Non-Standard Evolution

In this section, I discuss two possible features of NS evolution: magnetic field decay and fallback. Simplified tracks are shown in Figure 2. In addition, I present several examples of sources that absolutely do not fit the simplified NS evolution but that require either field decay, fallback, or both.

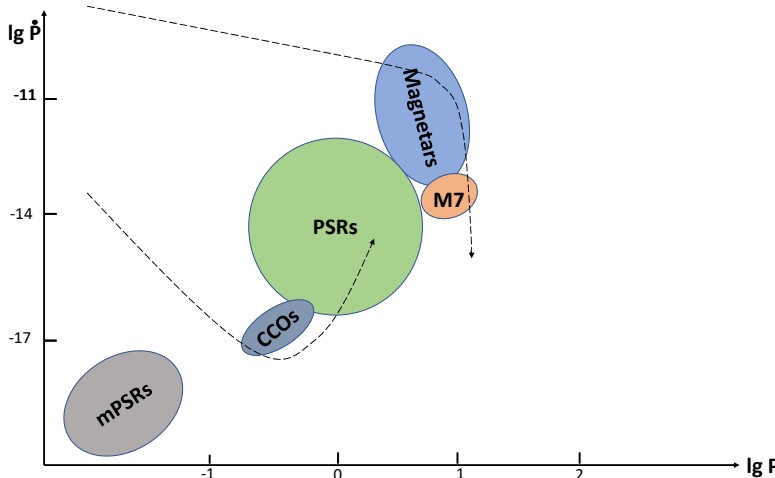

**Figure 2.** The same as in Figure 1, but with evolutionary tracks corresponding to field decay in magnetars and re-emerging fields in CCOs.

### 4.1. Magnetic Field Decay

It is quite natural to expect that magnetic fields of NSs might decay in time. In addition to obvious arguments about general physics that electric currents that are responsible for the magnetic field might decay due to finite conductivity in the crust (and maybe due to other processes, such as transport of magnetic flux tubes from the core to the crust, e.g., [91]), there are observational arguments. The existence of low fields in old NSs was demonstrated, e.g., through the discovery of mPSRs [5].

The theory of magnetic field decay in NSs and related observational data have been reviewed many times, see e.g., [64,65] and references therein. Here, I briefly mention several basic features.

A magnetic field can exist in the solid NS crust, in the liquid (and, most probably, superconducting) core, or in both. In the first case, the magnetic field is produced by electric currents. In the second, the field is confined in magnetic flux tubes, as the core is expected to be a type II superconductor, see e.g., [92]. Thus, the physics of the field evolution is very different in these two cases.

The physics of the core is much less understood, partly because only the field evolution in the crust is considered. The basics of crustal field evolution are perfectly described in [93]. The two main time scales can be defined as the Ohmic ($\tau_{\text{Ohm}}$) and the Hall ($\tau_{\text{Hall}}$).

The Ohmic scale can be written as:

$$\tau_{\text{Ohm}} = \frac{4\pi\sigma L^2}{c^2}. \tag{3}$$

Here, $\sigma$ is conductivity, and $L$ is a length scale in the crust.

The Ohmic time scale depends on the temperature of the crust, as electrons can scatter off phonons, as well as on crustal composition (impurities). For usual field configurations, $L$ is sufficiently large enough ($\sim$a few hundred meters in deeper layers, comparable to the crust thickness; see [93]) to make the time scale relatively long. In young hot NSs, $\tau_{\text{Ohm}}$ can be about $10^5$ yrs; see, e.g., Figure 4 in [94]. In older cold NSs, this time scale is long, at $\sim 10^9$ yrs.

The rapid release of magnetic energy can proceed via the Hall cascade [95]. This is a non-dissipative process, but it reconfigures the field such that $L$ decreases, such that the field can now decay faster. The time scale of this process is:

$$\tau_{\mathrm{Hall}} = \frac{4\pi e n_e L^2}{cB(t)} \tag{4}$$

where $e$ is the elementary charge, and $n_e$ is the concentration of electrons. For fields $\sim 10^{15}$ G, it can be as small as $\sim$100 yrs, e.g., [32]. However, the value of $\tau_{\mathrm{Hall}}$ is not well known, and it can be two orders of magnitude larger for the same field at $10^{15}$ G. It is widely accepted that magnetar activity is related to the Hall cascade in the crust of NSs.

The evolution of magnetic fields in the core is described in a more complicated way (see a brief review in Section 3.2 in [65]). Recently, Gusakov, Kantor, and Ofengeim developed a new approach to calculate field behavior in superconducting cores [96,97]. In particular, their results suggest that in the vicinity of the crust, the time scale can be as short as $\sim$100 yrs [98]. This is intriguing, as it potentially gives an opportunity to explain magnetar activity in the framework of the evolution of the core's field.

From an observational point of view, there are many arguments in favor of decaying fields in young NSs of different types. Magnetar activity provides evidence for field decay, as the magnetic energy is released in bursts and is responsible for the heating of the crust. The active lifetime of magnetars might be short, as it comes from the independent age estimates of these sources (SNR and kinematic ages, associations with clusters of young stars, etc.). However, bursts can also be produced by older NSs, but they are more seldom [99].

The thermal properties of NSs also provide arguments in favor of field decay. For example, Pons et al. [100] demonstrated that, typically, a high-B NS cannot have a low surface temperature due to additional heating related to the magnetic energy release in the crust.

Analysis of the properties of high-mass X-ray binaries (HMXBs) showed that the distribution of magnetic fields of NSs in these systems is compatible with models of crustal field evolution [101].

Finally, even for normal PSRs, some modeling favored a decaying field alongside their evolution; see, e.g., [102] and references therein. A different conclusion was made in [103]. These authors constructed a modified model of a so-called "pulsar current" [104,105] and concluded that a significant fraction of normal pulsars experiences an episode of field decay with a time scale of $\sim 4 \times 10^5$ yrs at ages of $\lesssim 10^6$ yrs. Later, this decay might be terminated. This points to Ohmic decay due to electron scattering off phonons, as this type of decay disappears when a NS becomes sufficiently cold. This happens at the ages of $\lesssim 10^6$ yrs even for low-mass objects. In the same framework, anomalous braking indices of PSRs can be explained as well [106].

To summarize, magnetic field decay in NSs is now a standard ingredient for modeling their evolution. Different modes of decay with different time scales are possible. Thus, presently, the situation is far from being clear, which is why observations of peculiar objects are important. I discuss these some of these observations in the following subsection.

### 4.2. Werewolves and Secret Agents

Until the beginning of this century, it was possible to attribute each young NS to some well-defined category (PSR, AXP, SGR, CCO, M7, etc.). In 2002, the discovery of SGR-like bursts from an AXP was announced [50]. That was the first, but not very prominent, example of "double nature". It was not so prominent because for some time, it had already been suspected that AXPs and SGRs form the same family of objects—magnetars. SGRs may be slightly younger and thus more active. However, later on, more pronounced examples of transitions from one subpopulation to the other were found. Here, I give some examples.

PSR 1846-0258 is observed only as an X-ray source; the radio beam does not point toward Earth. The NS has $B \sim 5 \times 10^{13}$ G and the largest rotational energy losses $\dot{E}_{\rm rot}$ among PSRs. A plerion and a SNR have been observed around the source. The characteristic age is $\sim$884 yrs. The spin period is $\sim$0.33 s. Thus, it resembled a Crab-like PSR, with an order-of-magnitude large field and an order-of-magnitude longer period. However, in 2008, magnetar-like activity was reported from this object [51,107]. The X-ray luminosity of the PSR significantly increased, and it started to produce SGR-like bursts. This came to be the first example of a radio pulsar becoming a magnetar.

Another example of a pulsar $\rightarrow$ magnetar transition is PSR 1622-4950. It has a $P = 4.3$ s and a large period derivative corresponding to $B \sim 3 \times 10^{14}$ G. Since its discovery, it has been suspected [108] that the source can be a magnetar in a quiescent state. In 2017, the source re-activated [109]. Its X-ray luminosity significantly increased; however, no bursts were detected.

I have already described a very peculiar source in the SNR RCW 103, which initially resembled an atypical CCO but then appeared to be an active magnetar. Its activity continued in 2016 with an outburst [110] that then decayed following the general scenario of crustal magnetic energy release.

PSR J1852+0040 in Kes 79 has also been previously mentioned. It is a candidate for "hidden" magnetars, i.e., a magnetar covered during a fallback episode such that only its crustal (but not magnetospheric) activity can be visible. This NS has a spin period of $\sim$0.1 s [111]. If we assume that the magnetar scale field was buried by fallback, then the initial episode has been very rapid such that the NS had no time to increase the spin period due to the interaction with the fallback flow (as what most probably happened in RCW 103). Then, the spin period of PSR J1852+0040 could be "frozen". Thus, the present-day value could be similar to the initial spin. This means that magnetar formation does not necessarily require rotation with a rate much smaller than 0.1 s [112]. It is tempting to say that a compact remnant of SN 1987A can also be a "hidden" magnetar [40], as its progenitor was a product of a coalescence of two massive stars in a binary system [113]. This coalescence could enhance the spin rate and magnetic field of the stellar core of the progenitor. The discovery of more "frozen" magnetars can shed light on their initial rotation rate and thus on the mechanism of magnetar formation.

After a fallback episode, the field is expected to diffuse out on a time scale of $\sim 10^3$–$10^5$ yrs [114]. While the field is diffusing out, its external structure can be changed, which can prevent the appearance of radio pulsar emissions [115]. Still, there is a possibility of observing a PSR on the stage of field re-emergence. Then, it might have a very non-standard track in the $P$–$\dot{P}$ diagram. Such objects are known. The most famous example is PSR 1734-3333 [116]. This is a standard PSR, but its period derivative is rapidly increasing. The rate corresponds to a braking index of $\sim$1, $\dot{P} \propto P^{2-n}$ (the standard Equation (1) corresponds to $n = 3$). With $P = 1.17$ s and a large period derivative $\dot{P} = 2.28 \times 10^{-12}$ s/s, it can after some time, e.g., $\sim$20–30 kyrs, enter the region of magnetars; see [116].

Finally, it is necessary to say a few words about low-field magnetars, which have also been mentioned above. As proposed in [117], these sources can form a significant subpopulation of relatively old magnetars. These sources might demonstrate very seldom activity and have low quiescent luminosities. Their existence points to the possibility of a quasistationary configuration of magnetic fields with a relatively low dipolar component. In the following subsection, I discuss a recently proposed stable field configuration—the Hall attractor.

### 4.3. Fallback and Hall Attractor

Examples of the peculiar sources described above suggest that NS field evolution can follow non-standard routes. In this subsection, I consider two important features of such evolution: fallback and the Hall attractor.

The idea that some fraction of matter ejected after a bounce in a SN explosion can later fallback onto the NS was proposed in the early 1970s (see a brief historical review in the

introductory section of [118]). In 1995, Muslimov and Page [119] suggested that fallback can significantly influence the external magnetic field of NSs and delay the switching-on of the radio pulsar emission mechanism.

The scenario of magnetic field submergence due to fallback became popular when it was applied to CCOs by Ho [114] and then by Vigano and Pons [120]. These authors demonstrated that for a realistic fallback amount ($\Delta M \sim 10^{-6} - 10^{-4}\,M_\odot$), the magnetic field can be significantly submerged and diffuses out on a time scale of $\sim 10^3$–$10^4$ yrs. This perfectly fits the properties of CCOs and explains why "evolved CCOs" are not observed as purely thermal emitters with $P \lesssim 1$ s, e.g., [90].

Bernal et al. presented 2D and 3D simulations of magnetic field submergence due to fallback [38]. They modeled the dynamics of interactions between falling matter and magnetic fields on the scale of $\lesssim 100$ ms for different fallback rates. The authors showed that for rates of $\dot{M} \gtrsim 10\,M_\odot\,\mathrm{yr}^{-1}$, total submergence happens, and for lower rates, the field is just partially submerged. Such rates are realistic at early stages of fallback; thus, in some fraction of young NSs, the external magnetic field can be smaller than the crustal field by several orders of magnitude.

Fallback can be prevented by activity of the central source. This possibility has been neglected in, e.g., [38], but later, it was studied by a group of Japanese researchers [121,122]. These authors attribute the diversity of young NSs mainly to different amounts of fallback. In particular, in [121], they define the criteria according to which, in a simplified 1D model for a given fallback rate, a NS becomes a CCO, a PSR, or a magnetar, depending on its spin and magnetic field. In [122], the same situation was studied in more detail with a numerical approach, but again in the 1D approximation and without accounting for instabilities.

Advanced fallback calculations were performed in a framework of SN explosion modeling [123]. A specific feature of this modeling is the motion of the NS relative to the ejecta. It was shown that this results in spin-up of a newborn NS due to fallback and spin–velocity alignment. In this framework, a NS's spin in mostly determined by the fallback (and not by the spin rate of the progenitor).

The influence of fallback on the spin of a newborn NS was also studied in [63]. In this scenario, a NS formed from a slowly rotating progenitor star can be spun-up significantly enough that conditions necessary for magnetar formation are fulfilled. Thus, in this model, fallback also produces rapidly rotating compact objects. An opposite situation is also possible in other scenarios.

The interaction between a fallback disc and magnetic field of a magnetar can result in significant spin-down of the NS. This possibility was recently analyzed in [44]. The disc can penetrate the light cylinder within a wide range of realistic fallback rates. Thus, the NS enters the propeller stage of magneto-rotational evolution. At this stage, a compact object can spin-down rapidly. Periods of $\sim 10^2$–$10^4$ s can be easily reached even within the lifetime of the SNR ($\lesssim 10^5$ yrs). This scenario is applicable for recently discovered long-period pulsars, which are discussed in the following section.

Long-spin periods can also be reached if the magnetic field remains large for a long time. This is possible if the Hall cascade in a magnetar crust is terminated or is at least significantly slowed down. Such a situation has been found numerically, and the stage was named "the Hall attractor" [124,125]. Later, it was confirmed in [126,127].

In their original paper [125], the authors obtained that the attractor is reached in $\lesssim 1$ Myr for initial fields of $\sim 10^{14}$ G. For larger fields, it is reached faster. At the attractor stage, the dipolar field is about $\exp(-3) \times B_0$, where $B_0$ is the initial field. The details significantly depend on the model, in particular, on the initial conditions (see a review of magnetic field evolution in NSs in [65]). The bottom line is the following: the rapid initial Hall evolution of large magnetic fields can be significantly slowed; this potentially allows for the existence of NSs with relatively large fields at ages of at least $\sim 1$ Myr (which is also important for the explanation of magnetar candidates in accreting binary systems [128]). In this case, such objects can reach relatively large spin periods due to standard losses. This can help to explain some sources discussed in the next section.

## 5. New Puzzle, New Tracks

Recent discoveries of long-spin period pulsars demand new non-trivial evolutionary tracks in comparison with those shown in Figure 2.

MeerKAT observations allowed Caleb et al. to discover a radio pulsar PSR J0901-4046 with a record-long spin period of 76 s [129]. With $\dot{P} = 2.25 \times 10^{-13}$ s s$^{-1}$, the source has the characteristic age of 5.3 Myr. The magneto-dipole field estimate provides a value of $1.3 \times 10^{14}$ G. In the standard scenario of magneto-rotational evolution, such a combination of parameters is impossible due to short initial spin periods and the rapid decay of large magnetic fields.

GLEAM-X J162759.5-523504.3 is even more exotic, with the period of pulsations of its radio emissions at $\sim$18 min [130]. This object was discovered with help from the Murchison Widefield Array. The period derivative has not been measured yet. This prevents robust determination of the source's nature. Still, it is probably a NS and not a WD; see discussion and references in [131]. Moreover, it might be a magnetar, as an upper limit on $\dot{P} \lesssim 10^{-9}$ s s$^{-1}$ provides that the observed luminosity is larger than the rotational energy losses. Thus, an additional source of energy is necessary, and it can be magnetic energy of the magnetar.

If $\dot{P}$ of GLEAM-X J162759.5-523504.3 is close to the upper limit, then the dipolar field is $\sim 3 \times 10^{16}$ G. Such values have been never observed. The characteristic age for such a field is $\tau_{\text{ch}} \sim 10^4$ yrs, which is significantly larger that the expected time scale of the Hall cascade for such huge fields. If $\dot{P} \sim 10^{12}$ s s$^{-1}$, then the field is $\sim 10^{15}$ G and $\tau_{\text{ch}} \gtrsim 10^7$ yrs. Again, such a combination is not a part of the standard scenario of NS evolution.

Possible solutions are related to physical processes discussed in the previous section. Either the real ages of both objects are much smaller than their characteristic ages due to large initial spin periods or the dipolar magnetic field in both cases could survive for a very long time, much longer than the initial time scale of the Hall cascade. In Figure 3, we show the tracks with large initial spin periods (3a,b,c). The initial short-dashed part of the tracks pointing toward PSR J0901-4046 (tracks 3b,c) and GLEAM-X J162759.5-523504.3 (track 3a) corresponds to a rapid spin-down from a short period, just after the core collapse, to longer periods, e.g., due to interactions with the fallback disc (of course, this spin-down does not proceed with a constant period derivative, and thus, we do not take this part of the tracks literally). For PSR J0901-4046, two variants of further evolution are shown: standard field decay and stalled decay.

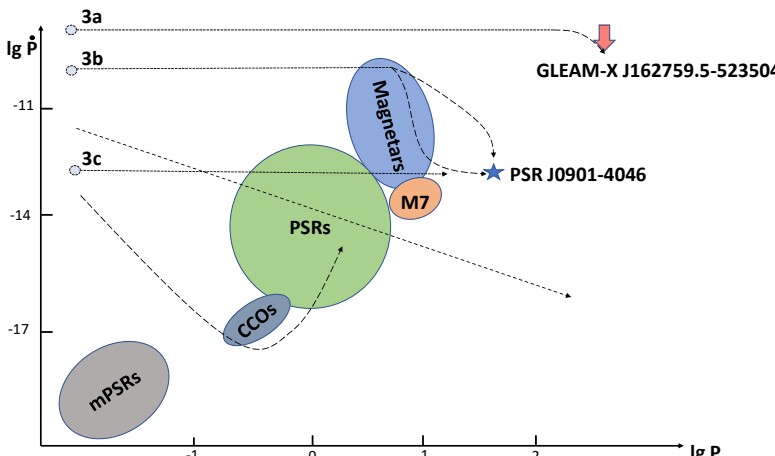

**Figure 3.** The same as in Figure 2, but with recently discovered long-period pulsars (the star symbol represents PSR J0901-4046, the arrow corresponds to the spin period and the upper limit on the period derivative of GLEAM-X 162759.5-523504) and tracks that can illustrate their evolution; see the text for details.

Note that above I discussed only the scenarios involving single stars. Evolution in a binary system can open an additional channel of producing long spin periods of NSs. If a NS in a HMXB system rapidly starts to accrete [132], or at least reaches the propeller stage, then its period can be rapidly increased up to hundreds or thousand of seconds in the case of large magnetic fields and accretion from a stellar wind; see a catalog of HMXBs in [133]. If this happens close to the moment of explosion of the secondary component, then we can expect a "birth" of an isolated NS with a large spin period.

NSs that can rapidly reach long spin periods (and that probably save a large value of their dipolar magnetic fields for a long time) can be of special interest for the long-term evolution of NSs. I discuss this in the following section.

## 6. Toward Accretion from the ISM

Already more than 50 years ago, it was suggested that isolated NSs can sooner or later start to accrete gas from the interstellar medium (ISM) [134,135]. More than 30 years ago, it was proposed that, e.g., ROSAT can detect thousands of accreting isolated NSs (AINSs) [136], but none have been found. This was explained in [137] as an evolutionary effect: most of isolated NSs under the standard assumptions cannot reach a stage of accretion during the lifetime of the galaxy. In addition, the rate of accretion onto the surface of a NS can be much lower than the standard Bondi value $\dot{M} \propto \eta \frac{(GM)^2}{v^3} \rho \sim$ $10^{11} \left( \frac{10\,\mathrm{km\,s^{-1}}}{v} \right)^3 \left( \frac{\rho}{10^{-24}\,\mathrm{g\,cm^{-3}}} \right) \mathrm{g\,s^{-1}}$ due to magnetic inhibition [138]. In the formula, $v$ is the NS velocity relative to the ISM, and $\rho$ is the ISM density, while the coefficient $\eta \sim 10$ depends on the details of the accretion flow around the NS.

Velocity distribution is an important ingredient of isolated NS evolution modeling, as interactions of a compact object with the interstellar medium strongly depend on this parameter. In addition, the initial velocity distribution determines the spatial distribution of NSs in the galaxy; see, e.g., [139]. Already, early observations have demonstrated that NSs can have spatial velocities that are significantly larger than their progenitors [140]. It is assumed that NSs obtain additional velocity at birth (so-called "kick"). The origin of kick, the shape of the velocity distribution, and the possible correlations of the kick velocity with other parameters are not yet completely understood. During last $\lesssim 50$ yrs, many attempt have been made to derive the kick velocity distribution from observations or to obtain it from theoretical considerations, e.g., SN explosion models (see a brief review and references to early studies in the introductory part of [141]). It is quite popular to use bimodal velocity distributions, as they fit better various data on radio pulsars and X-ray binaries (especially those with a Be-star donor). Recently, in [142], the authors presented a new analysis where they investigated the properties of radio pulsars and HMXBs. Their best fit was a bimodal distribution with $\sigma_1 \sim 30$–70 km s$^{-1}$ and $\sigma_2 = 336$ km s$^{-1}$, where 10–30% of NSs come from a low-velocity component. If isolated, such NSs can become potentially observable accreting sources within the galactic lifetime.

NSs with larger magnetic fields can start to accrete faster. This was studied in detail in [143]. The problem of low accretion luminosity can be solved in the settling accretion scenario [144]. In this framework, an accreting isolated NS can be observed as a relatively bright (e.g., using eROSITA) transient source [145]. However, the number of isolated accretors is still not expected to be very high, which makes these searches problematic.

The discovery of isolated accreting NSs is very much welcomed, as it can open a unique possibility to studying old isolated NSs and can help to learn a lot about their properties and evolution. eROSITA could be a perfect instrument to reach this goal [84]. On the other hand, it is important to provide better estimates of the number of accreting isolated NSs and their properties in order to simplify the identification of these objects.

In the standard approach [137], the main obstacle on the way to accretion is related to relatively slow spin-down of an isolated NS with a standard magnetic field of $\sim 10^{12}$ G. Recently, it was discovered that young, very long-period NSs are good candidates for reaching the stage of accretion in a relatively short time. Thus, an estimation of the number

of such objects is of great interest for long-term NS evolution. Illustrative evolutionary tracks for isolated NSs reaching the stage of accretion are shown in the Figure 4.

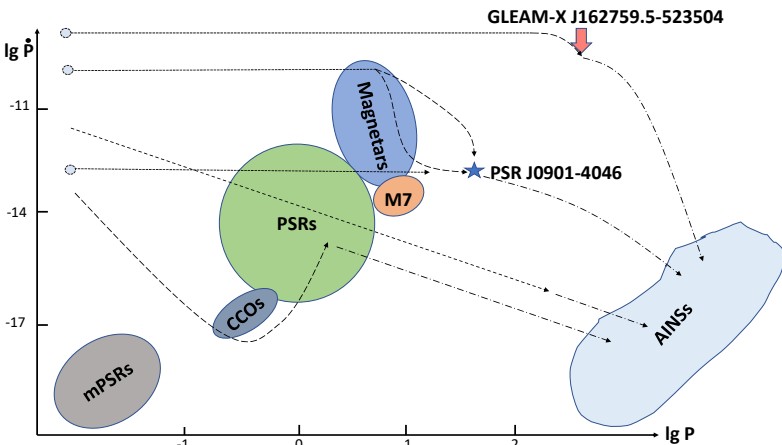

**Figure 4.** The same as in Figure 3, but with the addition of the region of AINSs and an illustration of the corresponding evolutionary tracks. See text for details. The position of AINSs is added out of scale.

In general, all NSs with long spin periods, large long-lived magnetic fields, or low spatial velocities (e.g., those born in $e^-$-captured SN) have good chances in reaching the stage of accretion from the ISM.

## 7. Magnetars and FRBs

Fast radio bursts (FRBs) are millisecond-scale radio transients that were discovered in 2007 [146]; see a recent comprehensive review in [147]. A possible link to NSs, in particular, to magnetars, was proposed in 2007 [148]. In 2020, it was confirmed through the detection of simultaneous radio and high-energy flares from the galactic magnetar, SGR 1935+2154 [149–154].

The number of known sources of FRBs is rapidly growing and is now at $\sim 10^3$. About 50 of the known sources demonstrate repeating activity [155], and four of the repeaters show a very high rate of events, producing up to several hundred bursts per hour [156]. In the near future, FRBs might become the most numerous known sources related to NSs, and they are extragalactic up to $z \gtrsim 1$. Thus, they will be one of the main sources of information regarding the universal population of NSs [157].

NSs that produce FRBs can have different peculiar properties and origin. It is expected that FRB sources are extreme magnetars with large fields that produce hyperflares with a total energy release of $\sim 10^{44}$ erg and peak luminosities of $\sim 10^{47}$ erg s$^{-1}$, which correspond to a millisecond radio burst with $L \sim 10^{43}$ erg s$^{-1}$ with a ratio of $\frac{L_{\rm radio}}{L_{\rm total}} \sim 10^{-4}$ (see, e.g., [153]).

Four of the most active FRB sources demonstrate such a huge rate of flares (hundreds per hour) that, from the energetic point of view, such behavior cannot last longer than a few years, as the whole magnetic energy of $\gtrsim 10^{47} \left( \frac{B}{10^{15}\,{\rm G}} \right)^2$ erg (see Equation (2)) would be emitted in this period [158]. Such intense outbursts are not observed among galactic magnetars.

Two of the repeating sources of FRBs demonstrate periodicity on the scale of $\sim 16$ [159] and $\sim 160$ [160] days. The origin of this periodicity is unknown. Among the proposed hypotheses, there are the following: binarity [161,162], NS precession [163,164], and extra-long spin periods [165]. All of these opportunities are very intriguing, as we do not have any robust examples of active magnetars in binary systems (see a review in [166]). We have only a few unconfirmed candidates for precessing magnetars (see, e.g., [167] and references therein), and we do not know any examples of such long spin periods of NSs.

The mission mechanism of FRBs has not been discovered yet. Presently, two main frameworks are discussed: magnetospheric emissions and external relativistic shocks; see reviews in [168,169]. A large number of advanced theoretical scenarios have been

proposed for both families of models. A growing variety of observational data (including polarization measurements, burst structure, spectra and their evolution during bursts) poses many questions and provides many opportunities to test these model predictions. The observations of simultaneous radio and X/$\gamma$-ray flares from galactic magnetars will probably help in selecting the correct approach. In general, understanding the origin of FRB radiation may shed light on important properties related to NS emission properties.

The galactic population of magnetars is consistent with the assumption that all these sources originated from core-collapsed SN. These NSs demonstrate clear correlations with young stellar populations and are sometimes situated inside standard SNRs; see, e.g., [170] for a review. However, a magnetar (or a NS, in general) can be formed via several other channels. Mostly, they are related to the coalescence of compact objects: NSs and/or WDs.

FRB sources are identified in different types of host galaxies in various environments [171], including in globular clusters [172]. The localization of FRBs at sites of very low star formation points toward alternative evolutionary channels related to old stellar populations. The coalescence NS-NS, NS-WD, and WD-WD altogether can produce NSs with a rate of at most $\sim 10^{-4}$ yrs$^{-1}$ per Milky-Way-like galaxy (see references in e.g., [157]). Thus, the probability of finding at least one active magnetar with such origins in our galaxy is not high. Observations of FRBs allow us to study these sources, even in different epochs of cosmic history. Moreover, in the near future, new sensitive low-frequency radio telescopes might allow us to observe FRBs from objects originating from Pop III stars.

Understanding the properties of the sources of FRBs can bring us new surprises regarding NS physics and observational appearances.

## 8. Conclusions

The field of NS astrophysics is actively developing due to the discoveries of new peculiar sources (such as long-spin period pulsars) and to the types of sources (such as FRBs). The phenomenology of NSs has become richer, and this requires more advanced theoretical approaches. We see more and more evolutionary links between the different beasts in the zoo of NSs. Understanding of this diverse population of sources is a fascinating task, and we must continue our research.

**Funding:** This research received no external funding.

**Acknowledgments:** I am grateful to the organizers of the 2nd International Electronic Conference on Universe (ECU 2023) and to Nicholas Chamel for the invitation to present a talk about different types of isolated NSs, which have become the basis for the present review.

**Conflicts of Interest:** The author declares no conflict of interest.

## Abbreviations

The following abbreviations are used in this manuscript:

| | |
|---|---|
| AINS | Accreting isolated neutron star |
| AXP | Anomalous X-ray pulsar |
| BH | Blackhole |
| CCO | Central compact object |
| FRB | Fast radio burst |
| HMXB | High-mass X-ray binary |
| ISM | Interstellar medium |
| LMXB | Low-mass X-ray binary |
| M7 | Magnificent seven |
| mPSR | Millisecond radio pulsar |
| NS | Neutron star |
| PSR | Radio pulsar |
| SGR | Soft gamma-ray repeater |
| sGRB | Short gamma-ray burst |
| SN | Supernova |

| SNR | Supernova remnant |
|-----|-------------------|
| WD | White dwarf |
| XDINS | X-ray, dim, isolated neutron star |

## Notes

1   https:www.atnf.csiro.au/people/pulsar/psrcat/, accessed on 30 May 2023.

2   See the on-line catalog at http://www.iasf-milano.inaf.it/~deluca/cco/main.htm, accessed on 30 May 2023. Mostly, the parameters of the CCOs mentioned in this subsection refer to this catalog.

3   http://www.physics.mcgill.ca/~pulsar/magnetar/main.html, accessed on 30 May 2023.

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
