# Peer review of "The Zoo of Isolated Neutron Stars"

_universe, doi:10.3390/universe9060273_

Round 1

Reviewer 1 Report

The author reviews the different populations of Neutron Stars (NS) and evolutionary paths in the P, P_dot diagram in relation with magnetic field decay and fallback matter. Phenomenoy is exposed in detail for some of these populations.

While I find the paper interesting some aspects could be improved. I list my main concerns below. 

-In the Introduction there is some discussion on binaries and detection of NS but there is no mention to the Nobel prize discovery. It truly deserves to be mentioned in this context.

-The current existing modeling behind some of the phenomenology is mentioned with random depth, for example, while magnetic field evolution in the core is explained and referenced, there is almost no mention to pioneering works of Baym et al, Horowitz et al about the crust and its impact in the evolution of NS. Some discussion should be added on pasta phases and crust.

-I find discussions regarding multimessengers for some NS related aspects should be at least mentioned, mergers and kilonovae will allow richer and more detailed characterizacion in the near future. Luminosity in SN is indeed key for additional fields, such as cosmological implications and current tensions  (Hubble tension) apart from NS phenomenology.

-Cooling NS is also worth being discussed in light of magnetized properties and populations. This is missing and some discussion would be indeed interesting to the reader.

-What about NS spatial and velocity  distribution in the galaxy for these NS populations? Only a few mentions are said about this. I find a paragraph is really needed. 

-Surveys and infrared telescopes will allow to detect some not so cold objects increasing the population samples. Can the author include a paragraph on this aspect?

-I find the reference list format is not appropriate with some entries displaying too many authors explicitly. Please, check usual conventions when editing. 

-English grammar and style is sometimes quite poor. A deep revision is needed.

-English grammar and style is sometimes quite poor. A deep revision is needed.

Author Response

I thank the referee for careful reading of the text and useful comments.
All of them are taken into account whenever possible.

Before answering to each comment of the referee I want to underline that the review is based in a 30-minute invited talk at the ECU 2023 conference organized by the Universe journal (MDPI): https://www.mdpi.com/journal/universe/events/14990. And the review is supposed to be published in a special issue which is a collection of materials presented at the conference (https://www.mdpi.com/journal/universe/special_issues/1HZ6F6C5T6). So, the review generally follows the line presented at the talk. I absolutely agree that much more information can be included, and each subject (pulsars, CCO, magnetars, field decay, fall back, ...) deserves a special review of the same size (or even more). Still, all this information (together with many additional references) hardly can be included in a review based on a 30-minute talk.

-In the Introduction there is some discussion on binaries and detection of NS but there is no mention to the Nobel prize discovery. It truly deserves to be mentioned in this context.

 I added the reference to Hewish et al. (1968) to sec. 2.1.  
In the Introduction there is a reference to the paper by Giacconi et al. (also, a Nobel result).

-The current existing modeling behind some of the phenomenology is mentioned with random depth, for example, while magnetic field evolution in the core is explained and referenced, there is almost no mention to pioneering works of Baym et al, Horowitz et al about the crust and its impact in the evolution of NS. Some discussion should be added on pasta phases and crust.

I agree that description of theoretical models sometimes is not complete.
I included some theoretical discussions of field decay and fall-back as they might be directly related to new discoveries - long period radio pulsars. Many other interesting theoretical issues (like cooling) are not discussed.

 Still, the reference to Baym et al. (1969) is added. 

As the review is mostly dedicated to phenomenology, I do not pretend to give a broad overview of theoretical models, especially of the history of understanding different subjects. 

-I find discussions regarding multimessengers for some NS related aspects should be at least mentioned, mergers and kilonovae will allow richer and more detailed characterizacion in the near future. Luminosity in SN is indeed key for additional fields, such as cosmological implications and current tensions  (Hubble tension) apart from NS phenomenology.

 There is a reference to a review on mergers, kilonova, etc. As this is not the main subject of the review (at the conference there were other review talks dedicated directly to the EoS, and I am sure that these topics will be covered by other authors - Chamel, Blaschke, Burgio, see the list of speakers at https://ecu2023.sciforum.net/#event_speakers)  I prefer to focus just on phenomenology of isolated NS. 

-Cooling NS is also worth being discussed in light of magnetized properties and populations. This is missing and some discussion would be indeed interesting to the reader.

 Generally, I agree. however, I had no time during the talk to touch this interesting and important subject. That is why it is not represented in the review. 
That is why I limited myself by refering to two reviews by Potekhin et al.
Still, in the revised version I tried to add some more words about cooling.

-What about NS spatial and velocity  distribution in the galaxy for these NS populations? Only a few mentions are said about this. I find a paragraph is really needed. 

 I agree. The paragraph is added. 

-Surveys and infrared telescopes will allow to detect some not so cold objects increasing the population samples. Can the author include a paragraph on this aspect?

 I disagree on this point. 
IR survey can be useful in looking for accreting isolated BHs. 
But in the case of INS I do not think that IR surveys can add something.

-I find the reference list format is not appropriate with some entries displaying too many authors explicitly. Please, check usual conventions when editing. 

 This is up to the people responsible for the journal style. I just use the journal template.

-English grammar and style is sometimes quite poor. A deep revision is needed.

 I tried to improve the text in this respect.

Reviewer 2 Report

Referee report to the paper “The zoo of neutron star” by Sergei Popov.

The author present a paper which pretending to be a short review of isolated neutron stars properties sand possible evolution connections between different types of isolated neutron stars. Unfortunately, the author is very biased in the references, he cited mainly own works or the works of his colleagues, and ignore a huge number of important works in the field. It means that the presented paper cannot be considered as a real review and has be significantly revised.

Remarks

1) The title is not completely correct. "The zoo of isolated neutron stars" would be more correct.

2) The author wrote when were discovered binary X-ray sources, but did not write when and how were discovered the various types of isolated neutron stars (INSs), namely pulsars, CCOs and so on, which are the main topic of the paper.

3) The author did not give description of the basic defining properties of the various
types of INSs at the beginning of the corresponding subsections.

4) Sect. 2.1 It is very strange that the author did not even mention the basic work of
Goldreich & Julian (1969), which established main principles of pulsar electrodynamic,  as well as very important work by Ruderman & Sutherlald (1975), and another fundamental work by Beskin et al. 1988. I would like also to recommend the review by Beskin (2018) as a very good introduction in the problems connected with the pulsars.

5) Sect. 2.2. The author did not mention pioneer works by Pavlov and his colleagues,
despite the fact that he is the person why invented this name, compact central objects
(CCO, Pavlov et al. 2000). In particular, after the sentence "These objects are observed ... with a thermal spectrum." references to the works by Pavlov and Zavlin, where this fact was established, are necessary.

6) Sect. 2.2. The work by Shabaltas & Lai (2012) was cited when CCO in Kes 79 is described. These authors were described the pulsed fraction value, but not the shape of the pulse profile. Therefore, the conclusions of this work cannot be considered as completely correct. Description of this CCO presented by Bogdanov (2014). The conclusions which were made in this paper could be also potentially interested for a given review.    

7) Sect. 2.3. It is very strange that the pioneer works by Thompson & Duncan (1993) was just briefly mentioned among other works describing the magnetic fields evolution in neutron stars. The next two fundamental works of the same authors about soft-gamma repeaters are not mentioned at all (Thompson & Duncan 1995, 1996). The contribution of such persons like Beloborodov or Kaspi were not also mention (see, e.g. the review by
Kaspi & Beloborodov 2017).

8) Sect. 4.1. The first line. Change "it" to "in".

9) Sect. 4.1. It is necessary shortly describe the physics of the Hall cascade. It is also strange that the pioneer work by Goldreich & Reisenegger (1992) did not mentioned.

10) page 9, line 370. "after some time enter.." Is it possible to give a quantitative estimation?

11) Sect. 4.3. page 10. The two paragraphs between lines 406 - 415 are in contradiction with the next paragraph. In the first paragraphs are described that the NS could be accelerated up to high spin rates due to fall-back accretion, but than the slow rotating NSs are also described using the same fall-back accretion if the strong NS magnetic field will be taken into account. But in the previous page (page 9, line 358) the fast rotating "hidden magnetar" in Kes 79 is also explained by fall-back accretion.  

12)page 10, line 434. The strong NS magnetic fields in accreting systems were suggested
earlier by Eksi et al. (2015) and Tsygankov et al. (2016).

13) Sect. 7. The review cannot be complete without mention of Beloborodov model
(Beloborodov 2020).

Author Response

I thank the referee for careful reading of the text and useful comments.
All of them are taken into account whenever possible.

Before answering to each comment of the referee I want to underline that the review is based in a 30-minute invited talk at the ECU 2023 conference organized by the Universe journal (MDPI): https://www.mdpi.com/journal/universe/events/14990. And the review is supposed to be published in a special issue which is a collection of materials presented at the conference (https://www.mdpi.com/journal/universe/special_issues/1HZ6F6C5T6). So, the review generally follows the line presented at the talk. I absolutely agree that much more information can be included, and each subject (pulsars, CCO, magnetars, field decay, fall back, ...) deserves a special review of the same size (or even more). Still, all this information (together with many additional references) hardly can be included in a review based on a 30-minute talk.

1) The title is not completely correct. "The zoo of isolated neutron stars" would be more correct.

I agree. Done.

2) The author wrote when were discovered binary X-ray sources, but did not write when and how were discovered the various types of isolated neutron stars (INSs), namely pulsars, CCOs and so on, which are the main topic of the paper.

Generally, I agree. I added a reference to Hewish et al. (1968) and Walter et al. (1996). 
References related to magnetar discovery were already in the paper. 
As for CCOs, it is not easy to say that there is a kind of a "discovery paper". So, in this case I prefer not to claim that there is a single distinguished discovery paper.

3) The author did not give description of the basic defining properties of the various
types of INSs at the beginning of the corresponding subsections. 

I agree. The text is modified accordingly. 
For the M7 (and partly for magnetars) the definition was already in the text. For all the rest I tried to add kind of definitions.

4) Sect. 2.1 It is very strange that the author did not even mention the basic work of
Goldreich & Julian (1969), which established main principles of pulsar electrodynamic,  as well as very important work by Ruderman & Sutherlald (1975), and another fundamental work by Beskin et al. 1988. I would like also to recommend the review by Beskin (2018) as a very good introduction in the problems connected with the pulsars.

Thank you for this comment. These papers are not mentioned as the review is mostly on observations, not on theory. I included some theoretical discussions of field decay and fall-back as they might be directly related to new discoveries - long period radio pulsars. Many other interesting theoretical issues (like cooling) are not discussed.

I absolutely agree that the review by V.S. Beskin is very informative, so I added a reference to it. 

5) Sect. 2.2. The author did not mention pioneer works by Pavlov and his colleagues, 
despite the fact that he is the person why invented this name, compact central objects
(CCO, Pavlov et al. 2000). In particular, after the sentence "These objects are observed ... with a thermal spectrum." references to the works by Pavlov and Zavlin, where this fact was established, are necessary.

I agree. The reference to Pavlov, Zavlin et  al. (2000) is added.

6) Sect. 2.2. The work by Shabaltas & Lai (2012) was cited when CCO in Kes 79 is described. These authors were described the pulsed fraction value, but not the shape of the pulse profile. Therefore, the conclusions of this work cannot be considered as completely correct. Description of this CCO presented by Bogdanov (2014). The conclusions which were made in this paper could be also potentially interested for a given review.    

The reference to Bogdanov (2014) is added.

7) Sect. 2.3. It is very strange that the pioneer works by Thompson & Duncan (1993) was just briefly mentioned among other works describing the magnetic fields evolution in neutron stars. The next two fundamental works of the same authors about soft-gamma repeaters are not mentioned at all (Thompson & Duncan 1995, 1996). The contribution of such persons like Beloborodov or Kaspi were not also mention (see, e.g. the review by Kaspi & Beloborodov 2017). 

The review is mainly on phenomenology, not on theory. That is why many references to theoretical studies are missing. Also, I want to underline that the review is based on a talk. So, I do not pretend to make the review much more informative and detailed than the talk was (of course, inevitably, the written review IS more detailed and containes much more references, still, I feel that it is necessary to keep correspondence with  contains of the talk).

8) Sect. 4.1. The first line. Change "it" to "in".

Done.

9) Sect. 4.1. It is necessary shortly describe the physics of the Hall cascade. It is also strange that the pioneer work by Goldreich & Reisenegger (1992) did not mentioned.

I agree. Corrected. 

10) page 9, line 370. "after some time enter.." Is it possible to give a quantitative estimation?

Yes, an estimate from Espinoza et al. (2011) is added.

11) Sect. 4.3. page 10. The two paragraphs between lines 406 - 415 are in contradiction with the next paragraph. In the first paragraphs are described that the NS could be accelerated up to high spin rates due to fall-back accretion, but than the slow rotating NSs are also described using the same fall-back accretion if the strong NS magnetic field will be taken into account. But in the previous page (page 9, line 358) the fast rotating "hidden magnetar" in Kes 79 is also explained by fall-back accretion.  

Yes. Different models predict different behaviour. It depends on parameters of fall-back. This stage is poorly understood, yet. It is quite natural to expect that fall-back in some cases can resul in spin-up, and in some - in spin-down. This is even underlined in the text by the sentence " An opposite situation is also possible in other scenarios." 
I tried to modify this part to make it more clear. 

12)page 10, line 434. The strong NS magnetic fields in accreting systems were suggested
earlier by Eksi et al. (2015) and Tsygankov et al. (2016). 

Here is some misunderstanding. In ref.[111] in the original manuscript (Igoshev, Popov 2018)
there was no PROPOSAL of high fields. This paper suggest a model to explain the existence of such fields.
And in the manuscript it corresponds to "rapid initial Hall evolution of large
 magnetic fields can be significantly slowed, this potentially allows existence of NSs with
relatively large fields ...". So, I believe that the reference is in place and it is not necessary to add other references. 
By the way, strong field in some accreting sources was proposed also long before the papers by Eksi and Tsygankov et al.  

13) Sect. 7. The review cannot be complete without mention of Beloborodov model
(Beloborodov 2020).

Indeed, this is an important paper. But I have to underline once again, that the review is not about theoretical models. No other theoretical models of FRB emission mechanism are discussed. So, only references to reviews are given, in this respect. 

Reviewer 3 Report

This manuscript provides a broad and comprehensive overview of the different classes of isolated neutron stars observed so far. Their characteristic properties are briefly described and the possible evolutionary links among some of them are discussed. I believe this paper could be of rather wide interest.

However, I have a number of comments listed below that the author should address before the paper can be accepted for publication.

- In the introduction, it would be worth mentioning that the recycling  scenario of neutron stars due to accretion from a stellar companion was indirectly supported by the discovery of the first millisecond pulsars and has been recently confirmed by the discovery of transitional millisecond pulsars, as recently reviewed e.g. by A. Papitto & D. d.  Martino in Bhattacharyya, S., Papitto, A., Bhattacharya, D. (eds) Millisecond Pulsars. Astrophysics and Space Science Library, vol 465  (Springer, Cham, 2022).

- In section 2.1, some words of caution should be added regarding the modelling of neutron stars as rotating magnetic dipoles in vacuum. This model is not expected to grasp the full complexity of the electrodynamics of neutron stars.  See, e.g., J. Pétri, Monthly Notices of the Royal Astronomical Society, Volume 485, Issue 4, p.4573-4587 (2019).

- Line 62, please provide references to RRATs.

- Line 83, it would be worth adding a few words about other observatories such as FAST.

- The figures would be much more informative if typical estimates for the period and its derivative were provided. Details on the evolutionary tracks should be also given.  What do the star and arrow mean in figures 3 and 4?  Please explain.

- Line 242, the author wrote: "Sometimes, these initial spin periods are assumed to be close to the initial period of the Crab pulsar. Such assumptions were very popular, for example, in binary population synthesis". When was this assumption dropped and why?

- Line 266, shouldn't the word "Despite" be replaced by "Apart from" or "In addition"?

- Line 277, references should be added to support the statement that "the core is expected to be type-II superconductor".

- Line 281, the sentence "The crustal field evolution is perfectly described in Ref.[76]" is a very strong statement, suggesting that no progress has been made since 2004. Please clarify. 

- Line 286, the author wrote: "For usual field configurations L is sufficiently large". Compared to what? Please explain.

- Line 299, the author wrote: "In the first place, magnetar activity provides
evidence for the field decay, as obviously the magnetic energy is released in bursts and it is responsible for the crust heating". The magnetic field decay might only be indirectly responsible for the crust heating through induced nuclear processes, as discussed by N. Chamel, A.F.  Fantina, L. Suleiman, J.-L.  Zdunik, P. Haensel, Universe 2021, 7(6), 193.

- Line 357, regarding the discussion about SN1987A the author could mention the observational evidence of a compact source as reported by P. Cigan, M. Matsuura, H.L. Gomez et al., The Astrophysical Journal, Volume 886, Issue 1, id.51 (2019). See also the discussion from D. Page, M. V. Beznogov, I I. Garibay et al., The Astrophysical Journal, Volume 898, Issue 2, id.125  (2020). 

- Line 395, the accretion rate seems wrong. From the value given in Ref.[23], I find that the rate is of order 1E-8 solar masses per year.  Please check.

Author Response

I thank the referee for careful reading of the text and useful comments.
All of them are taken into account whenever possible.

Before answering to each comment of the referee I want to underline that the review is based in a 30-minute invited talk at the ECU 2023 conference organized by the Universe journal (MDPI): https://www.mdpi.com/journal/universe/events/14990. And the review is supposed to be published in a special issue which is a collection of materials presented at the conference (https://www.mdpi.com/journal/universe/special_issues/1HZ6F6C5T6). So, the review generally follows the line presented at the talk. I absolutely agree that much more information can be included, and each subject (pulsars, CCO, magnetars, field decay, fall back, ...) deserves a special review of the same size (or even more). Still, all this information (together with many additional references) hardly can be included in a review based on a 30-minute talk.

- In the introduction, it would be worth mentioning that the recycling  scenario of neutron stars due to accretion from a stellar companion was indirectly supported by the discovery of the first millisecond pulsars and has been recently confirmed by the discovery of transitional millisecond pulsars, as recently reviewed e.g. by A. Papitto & D. d.  Martino in Bhattacharyya, S., Papitto, A., Bhattacharya, D. (eds) Millisecond Pulsars. Astrophysics and Space Science Library, vol 465  (Springer, Cham, 2022).

 I added in the Introduction a reference to the discovery paper (Backer et al. 1982),
and to Papitto et al. (2013) - the first transitional mPSR discovery.

- In section 2.1, some words of caution should be added regarding the modelling of neutron stars as rotating magnetic dipoles in vacuum. This model is not expected to grasp the full complexity of the electrodynamics of neutron stars.  See, e.g., J. Pétri, Monthly Notices of the Royal Astronomical Society, Volume 485, Issue 4, p.4573-4587 (2019).

I agree. The reference and warning are added. 

- Line 62, please provide references to RRATs.

Done.

- Line 83, it would be worth adding a few words about other observatories such as FAST.

References to results of FAST observations are added.

- The figures would be much more informative if typical estimates for the period and its derivative were provided. Details on the evolutionary tracks should be also given.  What do the star and arrow mean in figures 3 and 4?  Please explain.

Captions of FIgs. 3 and 4 are modified. Figures are modified, too.

- Line 242, the author wrote: "Sometimes, these initial spin periods are assumed to be close to the initial period of the Crab pulsar. Such assumptions were very popular, for example, in binary population synthesis". When was this assumption dropped and why?

It is difficult to define a particular moment when more advanced assumptions started to be used. 
It was related to gradual progress in our understanding of initial parameters of NSs (which are still understood not well enough). I tried to modify the text.

- Line 266, shouldn't the word "Despite" be replaced by "Apart from" or "In addition"?

I agree. Changes to "In addition to".

- Line 277, references should be added to support the statement that "the core is expected to be type-II superconductor".

Reference to the original paper by Baym et al. (1969) is added.

- Line 281, the sentence "The crustal field evolution is perfectly described in Ref.[76]" is a very strong statement, suggesting that no progress has been made since 2004. Please clarify. 

I agree. I changed it to: "Basics of the crustal field evolution are ...."

- Line 286, the author wrote: "For usual field configurations L is sufficiently large". Compared to what? Please explain.

The paragraph is modified.

- Line 299, the author wrote: "In the first place, magnetar activity provides
evidence for the field decay, as obviously the magnetic energy is released in bursts and it is responsible for the crust heating". The magnetic field decay might only be indirectly responsible for the crust heating through induced nuclear processes, as discussed by N. Chamel, A.F.  Fantina, L. Suleiman, J.-L.  Zdunik, P. Haensel, Universe 2021, 7(6), 193.

I agree that there is such a possibility. Still, at the moment this is not clear. In more popular scenarios crustal heating is directly due to magnetic energy release (i.e., current dissipation). 
As the review is mostly on phenomenology, I prefer to avoid discussions of different theoretical possibilities. 

- Line 357, regarding the discussion about SN1987A the author could mention the observational evidence of a compact source as reported by P. Cigan, M. Matsuura, H.L. Gomez et al., The Astrophysical Journal, Volume 886, Issue 1, id.51 (2019). See also the discussion from D. Page, M. V. Beznogov, I I. Garibay et al., The Astrophysical Journal, Volume 898, Issue 2, id.125  (2020). 

The evidence proposed my Cigan et al. is very indirect. Thus, a broad discussion is necessary which is not appropriate, in my opinion, in a compact review.

- Line 395, the accretion rate seems wrong. From the value given in Ref.[23], I find that the rate is of order 1E-8 solar masses per year.  Please check.

In Bernal et al. (2013) the authors state in the abstract: "In our simulations
we find the transition from total to partial submergence to occur around  \dot M ∼ 10 Msun yr−1".
https://iopscience.iop.org/article/10.1088/0004-637X/770/2/106/pdf
So, I quote this value as the one corresponding to complete submergence of the field.

Reviewer 4 Report

This article is an excellent review including recent discovery of long period neutron stars and fast radio bursts and is interesting for many readers. I therefore recommend the publication. Here are some minor comments to improve the draft.

(1)Location of Fig.1

Figure1 is cited on page 2(sec.2) and on page 5(sec.3), but appears on page 6. It is better to move it on page 2-5.

(2) I.e, E.g,

On Lines 118, 132 and 305, these are used in the top sentence. It is better to replace them “That is”, “For example”.

(3) “AINSs” in Fig.4

“AINSs” is not explained even in Abbreviations section (on Line 572).

Author Response

I thank the referee for careful reading of the text and useful comments.
All of them are taken into account.

>(1)Location of Fig.1
>Figure1 is cited on page 2(sec.2) and on page 5(sec.3), but appears on page 6. It is better to move it >on page 2-5.

The Figure is moved.

>(2) I.e, E.g,
>On Lines 118, 132 and 305, these are used in the top sentence. It is better to replace them “That is”, >“For example”.

Corrected,

>(3) “AINSs” in Fig.4
>“AINSs” is not explained even in Abbreviations section (on Line 572).

Absolutely correct. I added the abbreviation into the text and to the table of abbreviations. 

Round 2

Reviewer 3 Report

The author has satisfactorily revised his manuscript.